# Contesting the presence of wheat in the British Isles 8,000 years ago by assessing ancient DNA authenticity from low-coverage data

**Clemens L Weiß[1], Michael Dannemann[2], Kay Prüfer[2], Hernán A Burbano[1]***

[1]Research Group for Ancient Genomics and Evolution, Department of Molecular Biology, Max Planck Institute for Developmental Biology, Tübingen, Germany; [2]Department of Evolutionary Genetics, Max Planck Institute for Evolutionary Anthropology, Leipzig, Germany

**Abstract** Contamination with exogenous DNA is a constant hazard to ancient DNA studies, since their validity greatly depend on the ancient origin of the retrieved sequences. Since contamination occurs sporadically, it is fundamental to show positive evidence for the authenticity of ancient DNA sequences even when preventive measures to avoid contamination are implemented. Recently the presence of wheat in the United Kingdom 8000 years before the present has been reported based on an analysis of sedimentary ancient DNA (Smith et al. 2015). Smith et al. did not present any positive evidence for the authenticity of their results due to the small number of sequencing reads that were confidently assigned to wheat. We developed a computational method that compares postmortem damage patterns of a test dataset with bona fide ancient and modern DNA. We applied this test to the putative wheat DNA and find that these reads are most likely not of ancient origin.

*For correspondence: hernan. burbano@tuebingen.mpg.de

**Competing interests:** The authors declare that no competing interests exist.

## Introduction

The evolutionary reconstruction of the past has been greatly enriched by direct interrogation of ancient DNA (aDNA) from plants and animal remains (*Shapiro and Hofreiter, 2014*). Although a vast proportion of flora and fauna do not fossilize, traces of their DNA may be preserved in sediments allowing the characterization of past biodiversity (*Pedersen et al., 2015*). A challenge to exploiting such resources is the ubiquitous threat of contamination with exogenous DNA. Therefore, special sample preparation procedures have been developed to reduce DNA contamination (*Cooper and Poinar, 2000*). Nevertheless, it remains difficult to estimate how well preventive measures work. If contamination is a possible explanation for the result, it is crucial to exclude this possibility by giving positive evidence for the authenticity of aDNA (*Prüfer and Meyer, 2015*). Fortunately, a large number of full-length DNA sequences can be generated using next generation sequencing, which allows for the authentication of aDNA. In aDNA an excess of C-to-T (cytosine to thymine) substitutions occur at the 5′ and 3′ ends of molecules (or its mirror image G-to-A (guanine to adenine) at the 3′ end, depending on the library protocol employed). When considering the 5′ end of sequences, the excess of C-to-T substitutions is highest at the first base and decreases exponentially towards the center (*Figure 1A*). This pattern is the result of cytosine deamination to uracil in single stranded overhangs (*Briggs et al., 2007*). Since it is present in aDNA-derived sequences but absent in much younger samples, it has been used as an authentication criterion in aDNA experiments (*Krause et al., 2010*; *Prüfer and Meyer, 2015*).

Smith et al. analyzed sediments from Bouldnor Cliff, a submerged archeological site in the United Kingdom, and suggested the presence of domesticated wheat 8000 years ago based on sedimentary

**eLife digest** Ancient DNA, that is to say DNA extracted from fossils and ancient remains, provides a window into the past lives of humans, animals and plants. But working with ancient DNA is challenging; DNA decomposes with time, and so ancient DNA is often fragmented, damaged and present in tiny quantities. Furthermore, ancient DNA is also easily contaminated by modern DNA from those handling it and its surroundings. Researchers have therefore developed special protocols for working with ancient DNA and tests for its contamination.

One approach used to check that DNA is of ancient origin identifies a pattern of damage that is specific to ancient DNA. This damage changes the building blocks that make up DNA, causing one (called cytosine or C) to be misread as another (thymine or T). This substitution occurs most frequently at the ends of ancient DNA molecules, and occurs less often along its length, forming a detectable and characteristic pattern of damage.

A common way to analyse ancient DNA is to sequence it and then compare the resulting sequences to the genomes of modern organisms to identify its origins. In a study published earlier in 2015, investigators sequenced the DNA present in sediments obtained from a submerged archaeological site off the coast of the Isle of Wight in the United Kingdom. This previous study identified some DNA fragments that matched sequences in the wheat genome. This led the investigators to conclude that wheat was present in the British Isles around 8000 years ago, some 2000 years earlier than previously thought.

However, possibly owing to the small number of fragments that were found, the previous study did not check if the damage pattern matched that expected for ancient DNA. Now, Weiß et al. have developed a new computational method that tests whether DNA shows a typically ancient, or typically modern, pattern of C-to-T substitutions. When this test was used to assess the wheat sequences that were previously claimed to have ancient origins, it revealed that their pattern of DNA damage did not fit statistically with those of ancient DNA.

Weiß et al.'s findings contest those of the earlier study, and suggest that the new statistical method could be used to authenticate ancient DNA even when the number of available sequences is low.

ancient DNA (sedaDNA). This is 2000 years earlier than expected based on archeological remains in the British Isles and 400 years earlier than in nearby European sites (reviewed in Smith et al.). Since Smith et al. did not find wheat pollen or archeological remains associated with wheat cultivation, they conclude that the wheat presence in Bouldnor Cliff was the result of trading.

In total they produced ~72 million Illumina reads, of which they robustly assigned 152 to wheat (*Triticum*), with dozens more (160 reads) to higher taxonomic ranks that include wheat. Smith et al. took state-of-the art preventive measures to avoid contamination and exercised great effort to ensure the accuracy and robustness of their phylogenetic assignments. The authors attempted to authenticate the aDNA molecules based on the expected excess of C-to-T substitutions, but because of the very small number of reads assigned to wheat, they failed to do so using standard approaches. As a result of that, the authors did not present any positive evidence for the ancient origin of their reads. Here we present an approach that compares the pattern of C-to-T substitutions in a set of test reads with the distributions of C-to-T substitutions in reads from known ancient- and modern-DNA and apply this approach to sedaDNA from Smith et al.

## Results and discussion

Although the excess of C-to-T substitutions at the 5′ end occurs at different magnitudes in samples of different ages, the exponential increase of substitutions towards the end is a ubiquitous pattern in aDNA studies (*Sawyer et al., 2012*). In order to score the presence of this pattern in various datasets, we fitted an exponential function and evaluated the goodness of fit by using a one-sided t-test to test for significant exponential decay. As expected, true aDNA libraries show significant goodness-of-fit p-values (*Figure 1A*), whereas non-significant goodness-of-fit p-values, neither decay nor growth, are observed in libraries derived from modern DNA (*Figure 1B*). A given C-to-T damage pattern plot can

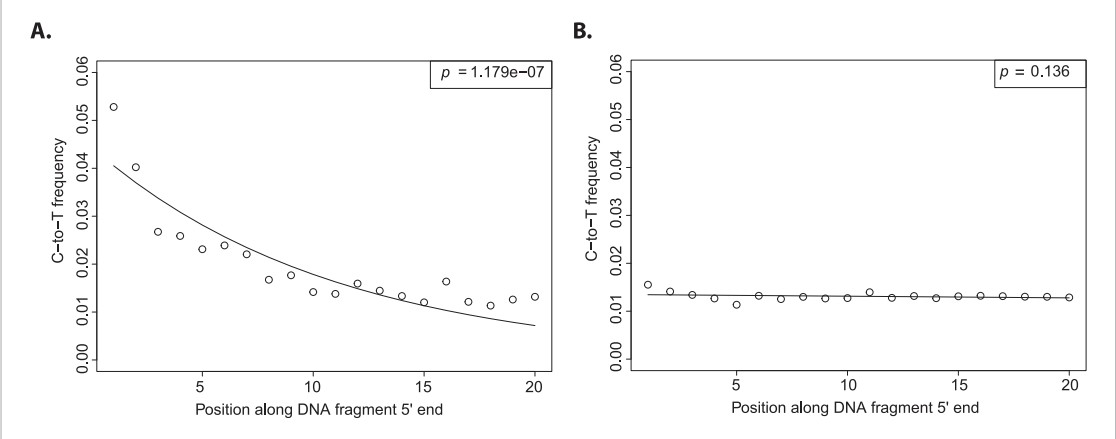

**Figure 1**. Patterns of cytosine to thymine (C-to-T) substitutions at the 5′end of known modern and ancient DNA. (**A**) C-to-T substitutions at the 5′ end from a whole library of historic *Solanum tuberosum* (ancient DNA). The line shows the fit with the exponential distribution and the box the goodness-of-fit p-value. (**B**) C-to-T substitutions at the 5′ end from a whole library of present-day *Triticum aestivum* (modern DNA). Line and box as in (**A**).

thus be summarized by its goodness-of-fit p-value that when it is significant indicates C-to-T exponential decay at the 5′ end (*Figure 1A*).

We resampled (with replacement) 10,000 sets of 150 sequences from a library of historic *Solanum tuberosum* collected in 1846 (*Yoshida et al., 2013a*). The number was selected to be comparable to the 152 reads that Smith et al. assigned to wheat. An empirical distribution of goodness-of-fit p-values was generated by performing the goodness-of-fit test for each subsample (*Figure 2A*). When we evaluate the sedaDNA goodness-of-fit p-value, we find that it falls within the upper 3% of subsamples with the least good fit. We can therefore reject the null hypothesis that the sequences assigned to wheat are as ancient as the historic *S. tuberosum* library. We repeated the whole procedure using this time a modern wheat library to generate the distribution of goodness-of-fit p-values (*Figure 2A*) and find a better match (p = 0.83). Thus, we cannot reject the hypothesis that the sequences assigned to wheat are of modern origin.

We sought to investigate how the test behaves when the empirical distribution of goodness-of-fit p-values is generated from different aDNA libraries. For this purpose we used a set of samples from animal (*Sawyer et al., 2012*) and plant remains (Yoshida et al., 2013) with an age of 85–170 years before present, and scored the sedaDNA wheat sequences against distributions generated from these libraries (subsamples of 150 sequences again). We observed that the goodness-of-fit p-value for the libraries is positively correlated with the empirical p-value for the sedaDNA wheat sequences tested against them (*Figure 2B*). Using a significance level of 0.05, we rejected the hypothesis that the wheat sequences are of ancient origin with 7 out of 13 libraries used in our test (*Figure 2B*). Thus, the purportedly 8000-year old wheat sequences show a less pronounced deamination pattern than many plant and animal samples with an age of less of 200 years. Finally, we took a less conservative approach and scored the sedaDNA against a distribution of goodness-of-fit p-values (subsamples of 150 read) generated from a 7000-years-old human Mesolithic sample from la Braña site in Northern Iberia (Olalde et al., 2014). La Braña is a site with cold environment and stable thermal conditions that has yielded exceptionally well conserved human fossils with ~50% of human endogenous DNA that reach a ~15% C-to-T substitution rate at the 5′ end (*Olalde et al., 2014a*) (*Figure 2—figure supplement 1*). We could reject the null hypothesis that the sedaDNA reads are as ancient as the sample from la Braña (p = 0.0014), a sample that is closer in time with the allegedly 8000-year-old wheat reads (*Figure 2—figure supplement 2*). It is worth pointing out that almost all 10,000 subsamples from la Braña had a very low (close to 0) goodness-of-fit p-value, even though we subsample only 150 reads (*Figure 2—figure supplement 2*).

We assessed the statistical power of the test by testing both an aDNA (*Figure 3A*) and a modern DNA library (*Figure 3B*) against a distribution built from a bona fide aDNA library, while varying the number of sampled sequences. Whereas the hypothesis that a true aDNA library is ancient was never

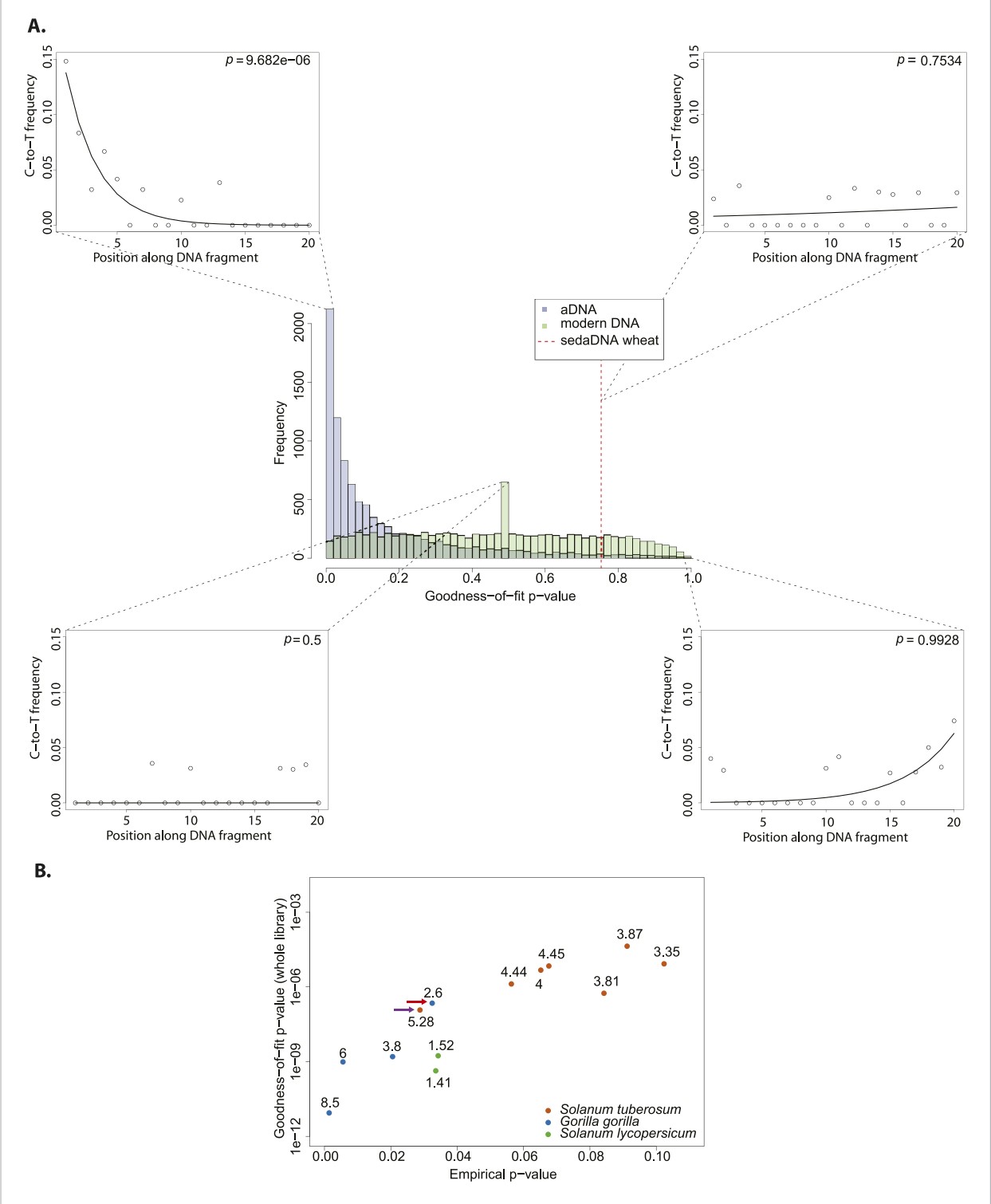

**Figure 2**. Authenticity test of DNA reads assigned to *Triticum* by Smith et al. (**A**) The histograms in the center panel show the empirical distributions of goodness-of-fit p-values of subsamples of 150 reads from ancient and modern DNA (same libraries as in *Figure 1*). The dotted red line indicates the location of the goodness-of-fit p-value from reads assigned to wheat in sedimentary ancient DNA. The four surrounding panels show cytosine to thymine (C-to-T) substitutions at the 5′ end extracted from different point of the goodness-of-fit p-value distributions, and from the reads assigned to wheat in sedimentary ancient DNA. (**B**) Variation of the empirical p-value of the test depending on the goodness-of-fit p-value of the whole library used to generate the empirical distribution. Numbers adjacent to the points indicate the percentage of C-to-T substitutions at first base. Red arrow indicates the aDNA library used as test in *Figure 3A*. Purple arrow indicates the library used to generate the empirical distribution of goodness-of-fit p-values in *Figure 3A–C*.

*Figure 2. continued on next page*

*Figure 2. Continued*

The following figure supplements are available for figure 2:

**Figure supplement 1**. Patterns of cytosine to thymine (C-to-T) substitutions at the 5′end from a 7.000-year-old Mesolithic human from La Braña site in Northern Iberia.

**Figure supplement 2**. Authenticity test of DNA reads assigned to *Triticum* by Smith et al.

rejected (*Figure 3A*), the hypothesis that a modern library has ancient origin could be rejected only when sufficient number of sequences were used for the subsample test (in tests with more than 300 reads the median empirical p-value was always below 0.05) (*Figure 3B*).

Finally, we skipped the phylogenetic curation step applied by Smith et al. to reduce the number of false positive wheat alignments, and mapped all reads sequenced by Smith et al. to the wheat genome. After stringent filtering of sedaDNA mappings we repeated our test varying the size of the subsample sets from 100 to 1000 reads. The empirical p-value was dependent on the number of reads tested, and declined with an increasing number of tested reads for all layers of sediments sequenced in Smith et al (*Figure 3C*). This pattern resembled the one obtained from a modern DNA library (*Figure 3B*). As for the phylogenetic curated 152 sequences, we were able to reject the hypothesis that the mapped reads are of ancient origin (mean p-value < 0.05 for all tests with more than 400 reads for layers 1–2 and 4, and 800 reads for layer 3). Our analysis also shows that the 152 sequences after phylogenetic curation are not a biased subsample from the distribution of all wheat-matching sequences.

We were able to reject the hypothesis that the sequences assigned to wheat by Smith et al. are of ancient origin. This is true even when we compared the putative 8000 year old sequences with only century old samples that show much lower deamination signatures. This means that a scenario in which wheat was transported to the Bouldnor Cliff site 8000 years ago is unwarranted. Our approach for authentication of aDNA can be used even with a very small number of sequences, and we hope that it will proof useful to test for positive evidence of authenticity for ancient DNA studies whose conclusions rely heavily on the ancient origin of the analyzed sequences.

## Materials and methods

### Read processing for bona fide ancient and modern DNA samples

Reads from most of the samples were downloaded from the European Nucleotide Archive (*Table 1* and *Supplementary file 1*), with the exception of the *Gorilla gorilla* reads that were provided directly by the authors (*Sawyer et al., 2012*). Adapters were trimmed for both paired- and single-end runs using the program Skewer (version 0.1.120) using default parameters (*Jiang et al., 2014*). For paired-end runs (*Supplementary file 1*) forward and reverse reads were merged requiring a minimum overlap of 10 base pairs (bp) using the program Flash (version 1.2.11) (*Magoc and Salzberg, 2011*). Merged or single-end reads were mapped as single-end reads against their respective nuclear or organellar genomes: *S. tuberosum* nuclear genome (*Potato Genome Sequencing Consortium et al., 2011*), *Solanum lycopersicum* nuclear genome (*The Tomato Genome Consortium, 2012*), *Triticum aestivum* nuclear genome (*International Wheat Genome Sequencing C, 2014*), *G. gorilla* mitochondrial genome (*Xu and Arnason, 1996*), *Homo sapiens* nuclear genome (Genome Reference Consortium Human Build 37). The mapping was carried out using BWA-MEM (version 0.7.10) with default parameters, which include a minimum read length of 30 bp (*Li, 2013*). PCR duplicates were removed after mapping using *bam-rmdup* (available at https://github.com/udo-stenzel/biohazard), which computes a consensus sequence for each cluster of duplicated sequences. Alignments were stored in the bam format (*Li et al., 2009*).

### Read processing for sedimentary DNA from *Smith et al., 2015a*

We used two different approaches to process the reads from sedimentary DNA (Smith et al., 2015).

Phylogenetic curated reads: we used a set of 152 reads assigned to tribe *Triticeae* and to genus *Triticum* by Smith et al. after phylogenetic curation. However, we consider the complete sequence

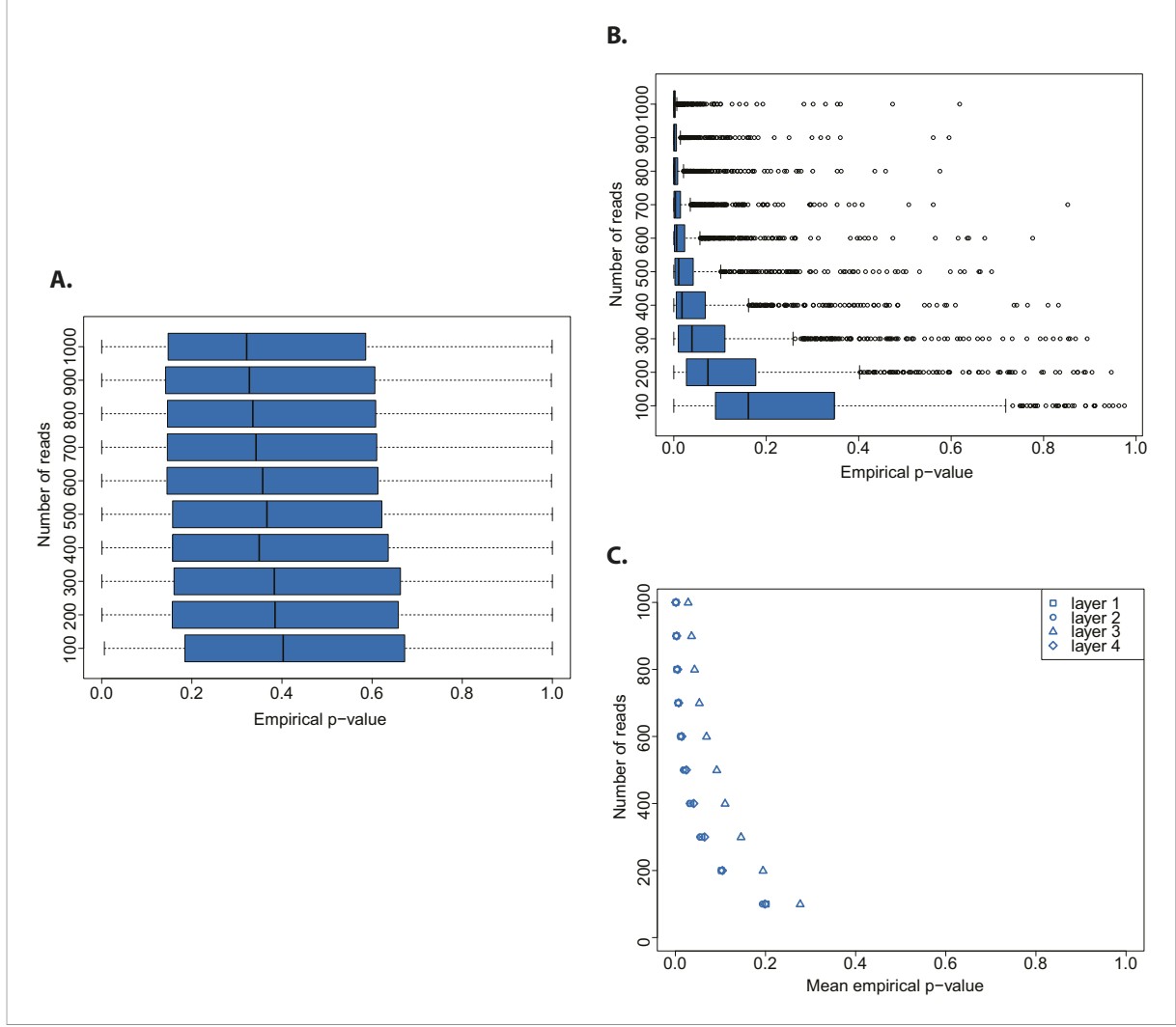

**Figure 3**. Evaluation of test performance. (**A**) Variation of the empirical p-value of the test depending on the number of reads sampled from an ancient DNA library (indicated with red arrow in **Figure 2B**). (**B**) Variation of the empirical p-value of the test depending on the numbers of reads subsampled from modern DNA *Triticum aestivum* library (same library used to generate the distribution of empirical goodness-of-fit p-values in **Figure 2A**). (**C**) Variation of the empirical p-value of the test depending on the size of sample sets from sedimentary ancient DNA reads mapped directly to the *T. aestivum* genome. Box-and-whisker plots were built based on 1000 tests. Layers as reported in Smith et al. i.e. layer 1 (most superficial), layer 4 (more deep).

and do not exclude the initial 10 nucleotides as was done in the original processing (Smith et al., 2015). Reads were then aligned to the wheat genome as described above.

All sedimentary DNA reads: we aligned independently reads from all four layers sequenced by Smith et al. to the *T. aestivum* nuclear genome (**International Wheat Genome Sequencing C, 2014**). Duplicates were removed and only alignments with mapping quality greater or equal than 30 were used for further analysis. Additionally, we include a sequence complexity filter based on entropy, which removed low complexity reads with entropy less or equal to 50. The entropy filtering was carried out with prinseq-lite (version 0.20.4) (**Schmieder and Edwards, 2011**).

## Exponential function fitting and calculation of goodness-of-fit p-value

For each set of aligned reads (complete libraries or subsamples) the C-to-T substitutions patterns along the 5′ end of the read were assessed using the program PMDtools (**Skoglund et al., 2014**). We fitted an exponential function to the frequency of C-to-T substitutions for the first 20 nucleotides at

**Table 1.** Provenance of samples

| Species | Type of DNA | Age | Reference | Study ID | Sample/run ID |
|---|---|---|---|---|---|
| Metagenomics sample | Sedimentary | 8030-7908* | Smith et al., 2015 | PRJEB6766‡ | ERR567364‡ |
| Metagenomics sample | Sedimentary | 8030-7908* | Smith et al., 2015 | PRJEB6766‡ | ERR567365‡ |
| Metagenomics sample | Sedimentary | 8030-7908* | Smith et al., 2015 | PRJEB6766‡ | ERR567366‡ |
| Metagenomics sample | Sedimentary | 8030-7908* | Smith et al., 2015 | PRJEB6766‡ | ERR567367‡ |
| Metagenomics sample | Sedimentary | 8030-7908* | Smith et al., 2015 | PRJEB6766‡ | ERR732642‡ |
| T. aestivum | Modern | NA | Chapman et al., 2015 | PRJNA250383‡ | SRR1170664‡ |
| S. tuberosum | Ancient | 135† | Yoshida et al., 2013 | PRJEB1877‡ | ERR267886‡ |
| S. tuberosum | Ancient | 137† | Yoshida et al., 2013 | PRJEB1877‡ | ERR267882‡ |
| S. tuberosum | Ancient | 149† | Yoshida et al., 2013 | PRJEB1877‡ | ERR330058‡ |
| S. tuberosum | Ancient | 165† | Yoshida et al., 2013 | PRJEB1877‡ | ERR267872‡ |
| S. tuberosum | Ancient | 166† | Yoshida et al., 2013 | PRJEB1877‡ | ERR267868‡ |
| S. tuberosum | Ancient | 166† | Yoshida et al., 2013 | PRJEB1877‡ | ERR957324‡ |
| S. tuberosum | Ancient | 167† | Yoshida et al., 2013 | PRJEB1877‡ | ERR267868‡ |
| S. lycopersicum | Ancient | 136† | Yoshida et al., 2013 | PRJEB1877‡ | ERR267884‡ |
| S. lycopersicum | Ancient | 139† | Yoshida et al., 2013 | PRJEB1877‡ | ERR267878‡ |
| G. gorilla | Ancient | 83† | Sawyer et al., 2012 | NA | 107¶ |
| G. gorilla | Ancient | 100† | Sawyer et al., 2012 | NA | 109¶ |
| G. gorilla | Ancient | 100† | Sawyer et al., 2012 | NA | 110¶ |
| G. gorilla | Ancient | 103† | Sawyer et al., 2012 | NA | 114¶ |
| Homo sapiens | Ancient | 7000* | Olalde et al., 2014 | PRJNA230689‡ | SRR1045127 |

*B.P. (before present years).
†Calculated from collection date (in years).
‡IDs from the European Nucleotide Archive.
¶IDs from **Sawyer et al., 2012**.

the 5′ end. The fitting was performed in R (http://www.r-project.org) using the *nls* function, which determines the nonlinear least squares estimates of the parameters in a nonlinear model. The fitting was carried out with the model formula: $y \sim N*\exp(-rate*x)$. From the *nls* fitting we obtained the t-value and degrees of freedom for the rate parameter and then calculated a goodness-of-fit p-value by using a one-sided t-test.

## Generation of empirical distributions of goodness-of-fit p-values

Subsets of different alignment numbers were randomly sampled (with replacement) 10,000 times from alignments stored in the bam format (*Li et al., 2009*). The random sampling was performed using samtools view (*Li et al., 2009*). For every subset of alignments we assessed the fraction of C-to-T substitutions, fitted an exponential function and calculated a goodness-of-fit p-value as explained above.

## Calculation of test empirical p-value

Phylogenetic curated reads: we compare the goodness-of-fit p-value of our test set of 152 sedimentary DNA reads with distributions of goodness-of-fit p-values generated from bona fide modern and ancient DNA. For the distribution of goodness-of-fit p-values from aDNA, we count how many of them are equal or greater than the sedimentary DNA goodness-of-fit p-value. To calculate the empirical p-value of the test we subsequently divided this number by the total number of values in the empirical distribution. With this approach we test the null hypothesis that the test set of reads contains a signal of ancient DNA damage that is comparable or even more pronounced than the signal in the aDNA library used to generate the empirical distribution of goodness-of-fit p-values.

For the distribution of goodness-of-fit p-values from modern DNA, we count how many of them were smaller or equal than the sedimentary DNA goodness-of-fit p-value. We calculate the empirical p-value of the test by dividing this number by the total number of p-values in the empirical distributions. With this approach we test the null hypothesis that the test set of reads matches the absence of ancient DNA damage patterns seen in reads of modern origin.

All sedimentary DNA reads: We tested independently alignments from each of the layers sequenced by Smith et al. using a bona fide aDNA sample for the generation of the distribution of goodness-of-fit p-values. For each layer we tested 10 sets of different numbers of reads (from 100 to 1000 reads, with increments of 100 reads). For each layer and for each number of reads in the test set we repeated the test and calculated the empirical p-value 1000 times as described above.

Other ancient DNA and modern DNA libraries were tested using the same procedure.

## Acknowledgements

We thank Moisés Expósito-Alonso, Janet Kelso, Rebecca Schwab and Detlef Weigel for discussions on data analysis and comments on the manuscript.

## Additional information

### Funding

| Funder | Grant reference | Author |
|---|---|---|
| Max-Planck-Gesellschaft (MPG) | | Michael Dannemann, Kay Prüfer |
| Deutsche Forschungsgemeinschaft (DFG) | SFB1052 | Michael Dannemann |
| Max-Planck-Gesellschaft (MPG) | Presidential Innovation Fund | Clemens L Weiß, Hernán A Burbano |

The funders had no role in study design, data collection and interpretation, or the decision to submit the work for publication.

### Author contributions

CLW, Conceived the project, developed statistical tests, analyze the data and wrote the paper; MD, Developed statistical tests; KP, Conceived the project and wrote the paper; HAB, Conceived the project, developed statistical tests and wrote the paper

### Author ORCIDs

Michael Dannemann, http://orcid.org/0000-0002-7076-8731

## Additional files

### Supplementary file

• Supplementary file 1. Sequencing strategy.

### Major datasets

The following previously published datasets were used:

| Author(s) | Year | Dataset title | Dataset ID and/or URL | Database, license, and accessibility information |
|---|---|---|---|---|
| Smith O, Momber G, Bates R, Garwood P, Fitch S, Pallen M, Gaffney V, Allaby RG | 2015 | Shotgun metagenomic study of sedimentary ancient DNA (sedaDNA), from four strata of sediment core taken from an Bouldner Cliff, a submarine archaeological site in the Solent. Dates of | http://www.ebi.ac.uk/ena/data/view/PRJEB6766 | Publicly available at the EBI European Nucleotide Archive (Accession no: PRJEB6766). |

| Author(s) | Year | Dataset title | Dataset ID and/or URL | Database, license, and accessibility information |
|---|---|---|---|---|
| Chapman JA, Mascher M, Buluç A, Barry K, Georganas E, Session A, Strnadova V, Jenkins J, Sehgal S, Oliker L, Schmutz J, Yelick KA, Scholz U, Waugh R, Poland JA, Muehlbauer GJ, Stein N, Rokhsar DS | 2015 | Triticum aestivum strain: SynOpDH Genome sequencing | http://www.ebi.ac.uk/ena/data/view/PRJNA250383 | Publicly available at the EBI European Nucleotide Archive (Accession no: PRJNA250383). |
| Yoshida K, Schuenemann VJ, Cano LM, Pais M, Mishra B, Sharma R, Lanz C, Martin FN, Kamoun S, Krause J, Thines M, Weigel D, Burbano HA | 2013 | Resequencing Solanaceae (Potato and Tomato) 19th century samples | http://www.ebi.ac.uk/ena/data/view/PRJEB1877 | Publicly available at the EBI European Nucleotide Archive (Accession no: PRJEB1877). |
| Sawyer S, Krause J, Guschanski K, Savolainen V, Pääbo S | 2012 | Temporal Patterns of Nucleotide Misincorporations and DNA Fragmentation in Ancient DNA | http://journals.plos.org/plosone/article?id=10.1371/journal.pone.0034131 | Publicly available at the Plos One (Accession no: 0034131). |
| Olalde I, Allentoft ME, Sanchez-Quinto F, Santpere G, Chiang CW, DeGiorgio M, Prado-Martinez J, Rodríguez JA, Rasmussen S, Quilez J, Ramírez O, Marigorta UM, Fernandez-Callejo M, Prada ME, Encinas JM, Nielsen R, Netea MG, Novembre J, Sturm RA, Sabeti P, Marques-Bonet T, Navarro A, Willerslev E, Lalueza-Fox C | 2014 | Illumina HiSeq 2000 sequencing; Whole genome sequencing of the La Brana 1 sample, AmpliTaqGold library | http://www.ebi.ac.uk/ena/data/view/SRR1045127 | Publicly available at the EBI European Nucleotide Archive (Accession no: SRR1045127). |

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
