## [Decision Letter]

Thank you for submitting your work entitled “Assessing ancient DNA authenticity with low-coverage data: a case study of wheat in the British Isles 8,000 years ago” for peer review at *eLife*. Your submission has been favorably evaluated by Mark McCarthy (Senior Editor) and three reviewers, one of whom is a member of our Board of Reviewing Editors.

The reviewers have discussed the reviews with one another and the Reviewing Editor has drafted this decision to help you prepare a revised submission.

The authors use an approach based on testing whether C-to-T damage patterns seen in degraded DNA fits a model of exponential decay from sequence termini in order to test a recent claim of 8,000 year old wheat DNA at a submerged site in Britain (Smith et al. 2015, Science).

Essential revisions:

The reviewers identified two additional analyses that they considered essential:

1) The authors should show visually the damage patterns in the reads from Smith et al.; even if this plot is noisy, it will provide some visual sense of the data.

2) The authors should provide some analysis of whether their results are robust to the choice of aligner. BWA-MEM uses extensive soft-clipping, which distorts patterns of damage. The authors should consider BWA-ALN/samse as an alternative.

Other reviewer comments below are provided as suggestions but need not be addressed in a revision.

Reviewer #1:

In this paper, the authors critically assess a claim from Smith et al. that wheat was present in the UK 8,000 years before present. Specifically, they argue that the sequencing reads claimed to come from ancient wheat samples in this study are instead most likely to be modern contaminants based on patterns of DNA damage.

1) The Smith et al. claim is based on 152 sequencing reads, and so the authors here have little to work with. However, they claim that the expected exponential increase in C->T substitutions in these reads is not present. It would have been nice to see the patterns of DNA degradation of these reads visually, rather than it being summarized by the p-value from their test. Why do the authors not show a figure like Figure 1, except using the 152 reads from Smith et al? This would be extremely useful for visually determining if the expected damage patterns are present (even if the plot is extremely noisy due to small numbers of reads). The authors could show the pattern from the empirical data superimposed on damage patterns in their subsampled ancient and modern libraries.

Reviewer #2:

Weiß et al. develop a pipeline to assess the authenticity of results of analyses of ancient sedimentary DNA, and use this pipeline to test recently published results. Such a pipeline is a useful tool for ancient sedimentary analyses, and is likely to be widely adopted in the field. I found the paper to be well written and straightforward. I have only one major query.

The authors choose to use BWA-MEM for alignment, which may not be the most appropriate aligner for this test and is likely to affect the results. This aligner (BWA-MEM) performs a lot of soft-clipping (unless the settings were modified from default, in which case the authors should state this), which causes two major issues. First, it makes misincorporation profiles unreliable. Second, it can cause a lot of random mapping. I strongly recommend the authors re-run these analyses using a more appropriate aligner, e.g. BWA-ALN/samse.

---

## [Author Response]

Essential revisions:

The reviewers identified two additional analyses that they considered essential:

1) The authors should show visually the damage patterns in the reads from Smith et al.; even if this plot is noisy, it will provide some visual sense of the data.

We thank the reviewers for the suggestion and agree that visualizing the damage patterns of the 152 reads assigned to *Triticum* by Smith et al. will facilitate the understanding of our analysis. We have modified Figure 1 and split it in two figures. In the new Figure 1 we have kept panels A and B, whereas in new Figure 2 we have original panels C and D. The new panel 2A now includes damage patterns from Smith et al. and also from different parts of the distributions of goodness-of-fit p-values of known ancient and modern DNA. We believe that the new figures provide an intuitive visual counterpart to different goodness-of-fit p-values.

2) The authors should provide some analysis of whether their results are robust to the choice of aligner. BWA-MEM uses extensive soft-clipping, which distorts patterns of damage. The authors should consider bwa aln/samse as an alternative.

We appreciate this very helpful comment and have performed a series of analyses to address it.

First we evaluate the amount of soft-clipping included in our BWA-MEM mapped reads. The percentage of mapped reads that are soft-clipped was very small for the reads assigned to *Triticum* in Smith et al. (5.4%), for the potato library reads used as known ancient DNA (aDNA) library (5.9%), and for the wheat library reads used as known modern DNA library (4%). Since our known aDNA library is affected by soft-clipping to a similar degree as the Smith et al. dataset, we do not expect a bias against an ancient signal. Nevertheless, we have repeated all analyses in an alternative approach using the software “PMDtools”, which handles soft-clipping by disregarding soft-clipped bases when estimating C-to-T damage patterns. Results were consistent using the two methods; in the manuscript we now report results based on “PMDtools”.

Second, we repeated the analysis by aligning all datasets with BWA-ALN using sensitive parameters that have been previously used for aDNA (no seed (–l 16500), allow more mismatches (–n 0.01) and allow up to two gap open events (–o 2)). Whereas we could map all 152 reads assigned to *Triticum* by Smith et al. to the *Triticum aestivum* reference genome using BWA-MEM, we could only map 78% of these reads (108 reads) using BWA-ALN (in Smith et al. reads were aligned using BLAST). We also aligned only a fraction of reads with BWA-ALN from the ancient potato (96%) and modern wheat libraries (73%). Despite this bias we found that the distribution of goodness-of-fit p-values from the known potato aDNA library with BWA-ALN resembled the distribution created with BWA-MEM (Figure 4). However, the wheat test dataset is not significantly different from the ancient potato reads (P=0.08) (Figure 4). This is most likely due to the reduced number of aligned reads; if we subsample 108 reads from the 152 reads in the original wheat dataset (10,000 mapping with BWA-MEM) and repeated the test the result is not significant in ∼48% of the cases.

Third, the result of the test also depends on the known aDNA library used to generate the empirical distribution of goodness-of-fit p-values. We took a rather conservative approach by using a *S. tuberosum* library that is only 169 year old, while the putative wheat reads from Smith et al. are claimed to be 8000 year old. Consequently, we have added a new analysis, where we used as known aDNA library from a 7.000-year-old Mesolithic human from La Braña site in Northern Iberia. We could again reject the null hypothesis that the test set of wheat reads are as ancient as the human fossil from La Braña (P=0.0014) (Figure 2—figure supplement 2). The distribution of goodness-of-fit p-values from La Braña shows that C-to-T signal is present in almost all subsamples of 150 alignments. The empirical p-values were significant independent on the choice of the aligner (Figure 5).

Author response image 1.Effect of aligner choice on the distribution of goodness-of-fit p-values and on the authenticity test.**(A)** Empirical distributions of goodness-of-fit p-values of subsamples of 150 reads from a 169-years-old *Solanum tuberosum* library. Reads were mapped using BWA-MEM and BWA-ALN. **(B)** Authenticity test of DNA reads assigned to *Triticum* by Smith et al. using same empirical distributions as in A. The dotted lines indicate the location of the goodness-of-fit p-value from reads assigned to wheat in sedimentary ancient DNA. The box encloses the empirical p-values of the authenticity tests.**DOI:**
http://dx.doi.org/10.7554/eLife.10005.012

Author response image 2.Effect of aligner choice on the distribution of goodness-of-fit p-values and on the authenticity test.**(A)** Empirical distributions of goodness-of-fit p-values of subsamples of 150 reads from a 7000-years-old human Mesolithic sample. Reads were mapped using BWA-MEM and BWA-ALN. To have a better visualization the y-axis is shown in a logarithmic scale. **(B)** Authenticity test of DNA reads assigned to *Triticum* by Smith et al. using same empirical distributions as in A. The dotted lines indicate the location of the goodness-of-fit p-value from reads assigned to wheat in sedimentary ancient DNA. The box enclosed the empirical p-values of the authenticity tests.**DOI:**
http://dx.doi.org/10.7554/eLife.10005.013